# Comparison between Robotic Single-Site and Laparoendoscopic Single-Site Hysterectomy: Multicentric Analysis of Surgical Outcomes

**DOI:** 10.3390/medicina59010122

**Published:** 2023-01-08

**Authors:** Barbara Gardella, Mattia Dominoni, Andrea Gritti, Liliana Mereu, Stefano Bogliolo, Marco Torella, Francesco Fanfani, Mario Malzoni, Aldina Couso, Alvaro Zapico, Ignacio Zapardiel

**Affiliations:** 1Department of Clinical, Surgical, Diagnostic and Paediatric Sciences, University of Pavia, 27100 Pavia, Italy; 2Department of Obstetrics and Gynecology, IRCCS Foundation Policlinico San Matteo, 27100 Pavia, Italy; 3Maternal and Child Department, Obstetrics and Gynecology, Cannizzaro Hospital, 95122 Catania, Italy; 4Department of Obstetrics and Gynecology, S Chiara Hospital, 38122 Trento, Italy; 5Department of Obstetrics and Gynecological Oncology, P.O del Tigullio Hospital-ASL4, Metropolitan Area of Genoa, 16043 Genoa, Italy; 6Obstetrics and Gynecology Unit, Department of Woman, Child and General and Specialized Surgery, University of Campania Luigi Vanvitelli, 80138 Naples, Italy; 7Gynecologic Oncology Unit, Department of Women, Children and Public Health Sciences, Fondazione Policlinico Universitario A. Gemelli IRCCS, 00168 Rome, Italy; 8Dipartimento Scienze della Vita e Sanità Pubblica, Università Cattolica del Sacro Cuore, 00168 Rome, Italy; 9Endoscopica Malzoni-Center for Advanced Endoscopic Gynecological Surgery, 83100 Avellino, Italy; 10Gynecology Department, Principe de Asturias University Hospital, 28805 Madrid, Spain; 11Gynecologic Oncology Unit, La Paz University Hospital, 28046 Madrid, Spain

**Keywords:** laparoscopy, minimally invasive surgery, hysterectomy, robotic surgery

## Abstract

*Background and Objectives*: Minimally invasive surgery, especially the single-site approach, has demonstrated several advantages in the gynaecological setting. The aim of this study was to compare the surgical outcomes of single-site hysterectomy for benign conditions between the traditional laparoendoscopic approach and robotic surgery. Materials and Methods: We consecutively enrolled 278 women between 2012 and 2019 in this multicentre trial. The patients underwent robotic single-site hysterectomy (RSSH) or laparoendoscopic single-site hysterectomy (LESSH) procedures with or without salpingo-oophorectomy for benign indications. Surgical parameters and surgical outcomes were analysed. Results: There was a statistical difference between the two surgical techniques for total operative time (*p* = 0.001), set-up time (*p* = 0.013), and anaesthesia time (*p* = 0.001). Significant differences in intraoperative blood loss were observed (*p* = 0.001), but no differences were shown for blood transfusion or intraoperative or postoperative complications in the two groups. Conclusions: LESSH outperformed RSSH in terms of surgical performance and clinical outcomes, with no differences in adverse events.

## 1. Introduction

Minimally invasive surgery, particularly the single-site approach (laparoscopic or robotics-assisted), has demonstrated several advantages in the treatment of benign and malignant conditions in the gynaecological setting. This surgical approach leads to well-known improvements over traditional laparotomy surgery, such as the absence of a large abdominal scar and a decrease in wound complications, postoperative pain, and hospital stay. Furthermore, the single-site technique improves aesthetic outcomes after surgery by reducing abdominal trauma and surgical incision scars, as well as the emergence of potential adverse effects (nerve, vessel, and tissue injuries) [1,2,3,4]. 

The published data underline how the transition from classic operative laparoscopy to the laparoscopic single-site technique caused the development of new surgical challenges as well as the interference of instruments, an unstable camera platform, a reduction in visualisation, reverse-handedness, and a loss of triangulation. The introduction of robotic technology permitted us to overcome some of these technical difficulties with an increase in surgical instrument performance and the well-known benefit of 3D visualisation [5]. Previous data on the benefits of laparoendoscopic single-site hysterectomy (LESSH) compared to robotic single-site hysterectomy (RSSH) have reported longer operative times for RSSH but higher reductions in estimated blood loss, shorter hospital stays, and shorter learning curves [6,7].

Analysing the timeline of the introduction of LESSH and RSSH, we found a gap of 41 years between the first reported surgery and the next ones to be reported. In 1969, Wheelss C. reported the first laparoscopic single-site procedure for tubal ligation [8], and in 2009, the first two LESSH procedures were reported by Langebrekke et al. [9] and Fader et al. [10]. In 2013, the Food and Drug Administration (FDA) approved the Da Vinci Robotics single-site platform. These chronological steps demonstrate the long-standing interest in minimally invasive techniques and the continuous improvement and development of the optics-, precision-, proficiency-, safety-, and ergonomics-related qualities of this surgical approach in order to decrease the insurgence of morbidity while increasing the well-being and quality of life of patients [11].

Nevertheless, to our knowledge, the published data on surgical comparisons between LESSH and RSSH are very limited and inconclusive, due to the confounding results reported [12]. On the other hand, a clear disadvantage of robotics with respect to laparoscopy is the higher cost of the equipment and the limited use of instruments [13]. However, as we reported in a previous paper, the robotic approach should be considered for selected patients in order to balance the advantages of robotic surgery with the costs of the technique with respect to the multiport approach [14].

The aim of the present study was to compare the surgical outcomes of single-site hysterectomy for benign conditions between the traditional laparoendoscopic approach and robotic surgery.

## 2. Materials and Methods

We carried out a multicentric retrospective study, including patients who underwent LESSH and RSSH with or without bilateral salpingo-oophorectomy (BSO) consecutively between March 2012 and December 2019, including the cases collected for the ULTRAMIS study [15]. The centres involved in the study were the following: IRCCS Policlinico San Matteo of Pavia, Italy; S Chiara Hospital, Trento, Italy; IRCCS University Hospital Foundation, Agostino Gemelli, Rome, Italy; Endoscopica Malzoni-Center for Advanced Endoscopic Gynecological Surgery, Avellino, Italy; Principe de Asturias University Hospital, Madrid, Spain; and La Paz University Hospital, Madrid, Spain.

The study was approved by the local institutional review boards of the participating centres, as well as that of the reference centre, the Local Institutional Review Board (reference n:7601/2017) of IRCCS Fondazione Policlinico San Matteo of Pavia, Italy.

All patients provided informed consent for the surgery (in accordance with the international law, i.e., the Declaration of Helsinki). All patients were informed at the time of providing consent that the LESSH and RSSH procedures could be converted to the multiport technique or a laparotomy if a surgical complication occurred during the procedure. All women were assigned to one of two groups: robotic single-site hysterectomy (RSSH) and laparoscopic single-site hysterectomy (LESSH).

Exclusion criteria for minimally invasive surgery included uterine size greater than 16 gestational weeks; a history of pelvic or abdominal radiation for a previous malignancy; any relevant disease that precluded a prolonged Trendelenburg position; severe hip diseases that precluded the dorso-lithotomy position; and, finally, any cervical or endometrial cancer requiring a radical hysterectomy and/or lymphadenectomy.

### 2.1. Surgical Procedures

The Da Vinci Si platform (Intuitive Surgical, Sunnyvale, CA, USA) was used in all robotic procedures by one qualified laparoscopic first surgeon, with two expert residents at the bedside and a dedicated surgical team. The LESSH was performed by one expert first surgeon (>50 laparoscopic hysterectomies per year) and two expert residents at the bedside.

The total operational time (TOt) was defined as the time between skin incision and skin closure. Setup time (ST) was defined as the time taken to set up the robot or laparoscopy. In both groups, the same technique was used for hysterectomy procedures and, eventually, BSO.

For the RSSH technique, a transumbilical 2.5 cm incision was made in the physiological umbilical hernia, and the single-site TM port, specific to the Da Vinci System SI (Intuitive Surgical, Sunnyvale, CA, USA), was used. The LESSH procedure was conducted using different devices, such as SILS (Coviden Medtronic, Madrid, Spain); Triport (Olympus Iberia, Barcelona, Spain); and Xcone (Karl Storz Endoskope, Tubingen, Germany), positioned in the umbilical scar. At the end of the hysterectomy procedure, the vaginal cuff was closed by a transvaginal approach for all procedures.

The estimated blood loss was defined as the drop in haemoglobin (Hb) value between the preoperative measurement and the first day after surgery. The same intraoperative anaesthesia and postoperative analgesia were provided to all women (ketorolac 30 mg twice per day and acetaminophen 1000 mg every 8 h). In both groups, a local infiltration of the abdominal fascia with 0.75% ropivacaine was executed at the time of surgical sutures. The visual analogue scale (VAS) was used in order to record postoperative pain at 1 h (T0) and 12 h (T1) from surgery, with an additional dose of analgesic drugs.

Any bladder, bowel, ureter, vessel, or nerve injury was defined as an intraoperative complication, as was an estimated blood loss (EBL) exceeding 500 mL. On the other hand, any adverse events that occurred six weeks after surgery were defined as postoperative complications, in agreement with the Clavien–Dindo classification [16]. The hospital stay was defined as the time from the day of surgery to the day of discharge. The first physical examination was performed 30 days after surgery in order to assess any postoperative adverse events.

### 2.2. Statistical Analysis

Descriptive statistics were computed for each case’s demographic, clinical, and laboratory characteristics. Mean and standard deviation (SD) are presented for normally distributed variables; median and interquartile range (IQR) for non-normally distributed variables; and number and percentages for categorical variables. Groups were compared with parametric or nonparametric tests, according to the data distribution, for continuous variables, and with Pearson’s chi-squared test (or Fisher’s exact test, where appropriate) for categorical variables. In all cases, two-tailed tests were used. The p-value significance cutoff was 0.05. The statistical software used was Stata (version 17; StatCorp, College Station, TX, USA).

## 3. Results

A total of 289 hysterectomies, with or without bilateral salpingo-oophorectomy for benign gynaecological diseases, were included in the study. Eleven patients were excluded due to concomitant surgical procedures performed (deep endometriosis that required bowel resection, urethral reimplantation, and a malignant surgical procedure that required lymph-node sampling). After this, laparoscopic single-site hysterectomies (LESSHs) were performed in 156 women and robotically assisted laparoscopic single-site hysterectomies (RSSHs) in 122 women. Table 1 summarises the demographic characteristics of the patients enrolled in the present study. There was a significant difference in the median age between the RSSH and LESSH groups (40.5 vs. 50.5 years, respectively; *p* = 0.001) and body mass index (BMI) (23 vs. 26 Kg/m^2^, respectively; *p* = 0.014). The median uterine weight (UW) in the RSSH group was slightly lower than that of the LESSH group but without statistical significance. However, when comparing the RSSH and LESSH groups, significant differences were observed in parity (nulliparity rate, 60.7% vs. 40.4%, respectively; *p* = 0.007); menopausal status (27.9% vs. 39.7%, respectively; *p* = 0.038); and operative indications (*p* < 0.001).

Table 2 reports the surgical parameters, operative times, hospital stay durations, and postoperative VAS values. There was a statistical difference between the two surgical techniques for TOt (RSSH median TOt 165 min vs. LESSH 120 min; *p* < 0.001) and ST (RSSH median ST 15 min vs. LESSH 10 min; *p* = 0.013). Finally, the anaesthesia time (AT) was higher in RSSH than in LESSH, with a statistical difference (195 min vs. 145 min, respectively; *p* < 0.001). Regarding the execution of salpingo-oophorectomy during the hysterectomy procedure, bilateral salpingo-oophorectomy was performed more frequently in the RSSH group than the LESSH group (89.34% vs. 62.18%, respectively; *p*-value < 0.001). Overall, the number of days of hospitalisation was greater in the RSSH group than the LESSH group (RSSH median 3.5 days vs. LESSH median 2 days; *p* = 0.001). At T0 (RSSH median VAS 5 vs. LESSH 1; *p* = 0.006) and T1 (median VAS 2 vs. 0, respectively; *p* < 0.001), pain assessment by VAS score revealed a statistical difference between the two groups. 

Table 3 shows the intraoperative and postoperative adverse events. Significant differences in intraoperative blood loss were observed (*p* < 0.001), but no differences were shown for blood transfusion or intraoperative or postoperative complications between the two groups. We noticed a significantly higher conversion rate in the RSSH group compared to the LESSH group (5.74% vs. 0.64%, respectively; *p* = 0.049). Significant differences in additional postoperative analgesia were observed between groups (*p* < 0.001). The most commonly used therapy in both groups was a combination of opioids and non-steroidal anti-inflammatory drugs (NSAIDs): 82.79% in the RSSH group and 54.49% in the LESSH group. Finally, the readmission rate was significantly higher in the RSSH group compared to the LESSH group (2.46% vs. 0%, respectively; *p* = 0.026).

## 4. Discussion

The single-site technique provides a theoretical advantage in gynaecologic surgery, but it is difficult to assess its differences from traditional minimally invasive surgery, and it is even more difficult to find differences between the robotic and laparoscopic approaches.

In a systematic review and meta-analysis, Albirighi et al. [12] showed that RSSH did not differ significantly from LESSH in terms of surgical outcomes in gynaecological but not oncological conditions; therefore, the authors declared that the safety and effectiveness of the robotic technique in comparison with laparoscopy was unclear, due to the non-significant differences in perioperative complications, length of stay, conversion to laparotomy, and blood loss. In this review, the outcomes of cost, pain, and quality of life were reported as inconsistent and not comparable. On the other hand, a review on the role of LESS conducted by Uppal et al. [17] reported that this technique was a possible approach in the gynaecological minimally invasive field, but its role was undetermined. The authors highlighted the advantages of single-site surgery, such as the increase in aesthetic outcomes and the reduction in postoperative pain; however, currently, the long-term results and possible benefits are not conclusive [18,19].

Analysing our results, the most important biases were the group differences regarding BMI, age, and uterine weight, because these depended on the distribution of patients included in our study. In the RSSH group, female-to-male sex reassignment was the main surgical indication; for this reason, we found younger, healthy, nulliparous, and normal-weight women in this group. This factor may have influenced the significant difference in demographic characteristics between the two groups.

Furthermore, BMI had no negative impact on surgical outcomes, and obesity did not appear to be a contraindication for a single-site approach; however, despite data supporting the use of robotic surgery in obese patients due to the benefits of this technique, we failed to demonstrate this advantage in our sample [20]. Age and BMI do not seem to influence the operative duration and outcomes of patients who undergo RSSH, in contrast with the surgical outcomes of other traditional types of surgery [21,22,23]. Gupta et al. reported that in younger patients with a lower BMI, a traditional laparoscopy or RSSH was more likely to be performed [24].

Therefore, in our sample, we excluded uterine volume > 16 weeks of gestation from the ergonomic limit of the single-site system, because this factor was an independent risk for postoperative complications. The single-site approach requires surgical experience, because the movements are forced and the vision is more limited, as literature data have underlined: “the reduction of operative time is related to the experience of surgeons, and surgical skill influences all operative times, despite the complexity of surgical cases” [9,10,11,12,13,14,15,16,17,18,19,20,21,22,23,24]. For laparoscopy, the learning curve requires that surgeons have substantial experience with traditional laparoscopy, while the robotic approach seems to be easier due to the magnified three-dimensional vision, wristed instruments, and improved dexterity. For this reason, as demonstrated in our previous research, the learning curve for the robotic single-site approach is feasible for younger surgeons [25].

Because the intraoperative adverse events necessitated an intracorporal bladder suture, conversion to laparotomy was more common for the robotic than the laparoscopic technique in our study. In the meta-analysis performed by Mereu et al. [26], the risk of the single-site laparoscopic conversion technique during hysterectomy was comparable to that of the multiport technique. The LESSH group had more postoperative adverse events than the RSSH group, but the total and operative time were longer in the RSSH group, because the most important limitation of this study was the different surgical skills represented without randomised enrolment. In addition, this study was a retrospective analysis of surgical outcomes, and a long follow-up of surgical and aesthetic outcomes was not available.

## 5. Conclusions

In conclusion, LESSH and RSSH showed no significant differences in surgical outcomes, but the total operative time, hospital stay duration, and hospital discharge time were higher in the RSSH group than in the LESSH group. In addition, the robotic approach was linked to a significant risk of intraoperative bleeding. We can reasonably conclude that LESSH has more benefits in terms of surgical performance and results, and that the cost of RSSH does not justify its use in this group of patients; however, we do acknowledge the easier learning curve in robotic surgery.

## Figures and Tables

**Table 1 medicina-59-00122-t001:** Demographic characteristics of patients enrolled.

*Characteristics*	*RSSH (n = 122)*	*LESSH (n = 156)*	*p-Value*
Age (years), median (IQR)	40.5 (28–52)	50.5 (36–54.50)	<0.001
BMI (kg/m^2^), median (IQR)	23 (21–27)	26 (23.25–28)	0.014
Uterine weight (g), median (IQR)	58.8 (46.3–111.20)	79 (67.50–107.50)	0.097
Nulliparous, n (%)	74 (60.66)	63 (40.38)	0.007
Menopausal status, n (%)	34 (27.87)	62 (39.74)	0.038
Surgical indication, n (%)			
Uterine fibromatosis	20 (16.39)	60 (38.46)	
Gender reassignment	68 (55.74)	30 (19.23)	
CIN3^+^	2 (1.64)	1 (0.64)	<0.001
Endometrial hyperplasia	32 (26.23)	65 (41.66)	

**Legend:** N = number of cases; IQR = interquartile range; g = grams; kg = kilograms; m = metres; BMI = body mass Index; RSSH = robotic single-site hysterectomy; LESS = laparoendoscopic single-site hysterectomy; CIN = cervical intraepithelial neoplasia.

**Table 2 medicina-59-00122-t002:** Surgical parameters in RSSH and LESSH groups.

*Characteristics*	*RSSH (n = 122)*	*LESSH (n = 156)*	*p-Value*
Salpingo-oophorectomy, n (%)			
Not performed	9 (7.38)	56 (35.9)	
Bilateral	109 (89.34)	97 (62.18)	
Monolateral	2 (1.64)	0 (-)	**<0.001**
Bilateral salpingectomy only	2 (1.64)	3 (1.92)	
Operation time (min), median (IQR)	165 (125–190)	120 (110–138.25)	**<0.001**
Setup time (min), median (IQR)	15 (10–25)	10 (10–15)	**0.013**
Anaesthesia time (min), median (IQR)	195 (160–226.25)	145 (125.00–159.25)	**<0.001**
Hospital discharge (days), median (IQR)	3.5 (3–4)	2 (2–3)	**0.001**
VAS T0, median (IQR)	5 (4–6)	1 (0–3)	**0.006**
VAS T1, median (IQR)	2 (1–4)	0 (0)	**<0.001**

Legend: VAS T0 = visual analogue scale 1 h after surgical procedure; VAS T1 = visual analogue scale 12 h after surgical procedure; RSSH = robotic single-site hysterectomy; LESSH = laparoendoscopic single-site hysterectomy.

**Table 3 medicina-59-00122-t003:** Surgical outcomes in RSSH and LESSH groups.

*Characteristics*	*RSSH* *N = 122*	*LESSH* *N = 156*	*p-Value*
Conversion to LPT, n (%)	2 (1.64)	0 (0)	** *0.049* **
Intraoperative complications, n (%)			
Bladder injury	3 (2.46)	2 (1.28)	
Bowel injury	0 (0)	0 (0)	
Vaginal laceration	1 (0.82)	1 (0.64)	0.135
Intraoperative bleeding, n (%)			
<50 mL	54 (44.26)	113 (72.43)	
50–200 mL	55 (45.08)	43 (27.56)	
200–500 mL	11 (9.01)	0 (0)	
>500 mL	2 (1.64)	0 (0)	** *<0.001* **
Intraoperative bleeding (mL, average ± SD)	(93.52 ± 127.84)	(42.85 ± 27.37)	** *<0.001* **
Blood transfusion, n (%)	1 (0.82)	1 (0.64)	0.862
Analgesia, n (%)			
Paracetamol	0 (0)	10 (6.41)	
NSAIDs	4 (3.28)	27 (17.31)	
Oppioid	9 (7.38)	30 (19.23)	** *<0.001* **
Oppioid + NSAIDs	109 (89.34)	89 (57.05)	
Postoperative complications, n (%)			
Vaginal suture dehiscence	1 (0.82)	1 (0.64)	
Vaginal haematoma	0 (0)	1 (0.64)	
Scar infection	0 (0)	1 (0.64)	
Bladder infection	1 (0.82)	1 (0.64)	
Trocar access hernia	0 (0)	2 (1.28)	0.159
Neuralgia	1 (0.82)	2 (1.28)	
Readmission, n(%)	3 (2.46)	0 (0)	** *0.026* **

**Legend:** RSSH = robotic single-site hysterectomy; LESSH = laparoendoscopic single-site hysterectomy; LPT = laparotomy; NSAIDs = non-steroidal anti-inflammatory drugs.

## Data Availability

The datasets generated and analyzed during the current study are available from the corresponding author on reasonable request.

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
