# Peer review of "Comparison between Robotic Single-Site and Laparoendoscopic Single-Site Hysterectomy: Multicentric Analysis of Surgical Outcomes"

_medicina, 2023, doi:10.3390/medicina59010122_

Round 1

Reviewer 1 Report

Dear authors,

I appreciated your manuscript, due to its managing and, especially due to its results. I am sure that the results are not comfortable for the institutions which invested a lot of money for robotic infrastructure. I liked the conclusion that if you want to pay more, you are free to do it, but this does not compulsory means you will receive more.

I consider that the manuscript can be published with small / minor completions:

The authors affirmed that it is a multicentric study (in the title and, more important) at line 76. I consider appropriate to be mention the names of the centers involved.

Also, during lines 79-80 it is specified that: “The study was approved by the Local Institutional Review Board (reference n: 7601/2017).”. This is only for one center, isn’t it? If it is true, the authors should provide the name of the IRB for all centers.

Author Response

1) The authors affirmed that it is a multicentric study (in the title and, more important) at line 76.  I consider appropriate to be mention the names of the centers involved.

The name of involved centers were added in the manuscript.

2)Also, during lines 79-80 it is specified that: “The study was approved by the Local Institutional Review Board (reference n: 7601/2017).”. This is only for one center, isn’t it? If it is true, the authors should provide the name of the IRB for all centers.

The approval of IRB for each center was included in the manuscript. 

Reviewer 2 Report

I think that this report is valid and interesting, because you started to use RSSH from 2012, probably it is very early. However, I will point out a little.

In table1, I think the patients with RSSH are young and slim. And the major surgical indication was gender reassignment, but gynecological diseases were fewer. This results seems that you selected easy cases for RSSH. However, total operative time, hospital stay and hospital discharge were higher in RSSH group. Therefore, I think your conclusion is valid.

However, I think that blood loss amount is one of the most important index. In table 2, you compare the median of operation time, but you did not compare the median of blood loss amount. You classified in 4 groups, including <50ml, 50-200ml, 200-500ml and >500ml. I want to know the median or average of blood loss amount in both groups, including RSSH and LESSH.

Author Response

1) In table1, I think the patients with RSSH are young and slim. And the major surgical indication was gender reassignment, but gynecological diseases were fewer. This results seems that you selected easy cases for RSSH. However, total operative time, hospital stay and hospital discharge were higher in RSSH group. Therefore, I think your conclusion is valid.

in consideration of multicentric retrospective analysis  there is a differences in  the surgical approach for each center, but, as the reviewer observed, the patients selected for RSSH was for the large parte gender reassignment.

2) However, I think that blood loss amount is one of the most important index. In table 2, you compare the median of operation time, but you did not compare the median of blood loss amount. You classified in 4 groups, including <50ml, 50-200ml, 200-500ml and >500ml. I want to know the median or average of blood loss amount in both groups, including RSSH and LESSH.

The data were added in table 2